# Complications and Mortality Rate of Cytoreductive Surgery with Hyperthermic Intraperitoneal Chemotherapy: Italian Peritoneal Surface Malignancies Oncoteam Results Analysis

**DOI:** 10.3390/cancers14235824

**Published:** 2022-11-25

**Authors:** Fabio Carboni, Mario Valle, Marco Vaira, Paolo Sammartino, Orietta Federici, Manuela Robella, Marcello Deraco, Massimo Framarini, Antonio Macrì, Cinzia Sassaroli, Piero Vincenzo Lippolis, Andrea Di Giorgio, Daniele Biacchi, Lorena Martin-Roman, Isabella Sperduti, Dario Baratti

**Affiliations:** 1Peritoneal Tumors Unit, IRCCS Regina Elena National Cancer Institute, 00144 Rome, Italy; 2Candiolo Cancer Institute, FPO—IRCCS, 10060 Candiolo, Italy; 3Cytoreductive Surgery and HIPEC Unit, Department of Surgery “Pietro Valdoni”, Policlinico Umberto I, Sapienza University of Rome, 00185 Rome, Italy; 4Peritoneal Surface Malignancy Unit, Department of Surgery, Fondazione IRCCS Istituto Nazionale Tumori, 20133 Milan, Italy; 5Surgery and Advanced Oncological Therapy, Morgagni—Pierantoni Hospital, AUSL Romagna, 47121 Forlì, Italy; 6UOC—PSG con OBI Azienda Ospedaliera Universitaria “G. Martino”, 98125 Messina, Italy; 7Colorectal Surgical Oncology, Abdominal Oncology Department, “Fondazione Giovanni Pascale” IRCCS, 80131 Naples, Italy; 8General and Peritoneal Surgery, Department of Surgery, Hospital University Pisa, 56126 Pisa, Italy; 9Surgical Unit of Peritoneum and Retroperitoneum, Fondazione Policlinico Universitario A. Gemelli IRCCS, 00168 Rome, Italy; 10Department of Biostatistical Unit, IRCCS Regina Elena National Cancer Institute, 00144 Rome, Italy

**Keywords:** peritoneal surface malignancies, cytoreductive surgery, HIPEC, morbidity, mortality, risk factors

## Abstract

**Simple Summary:**

Cytoreductive surgery with hyperthermic intraperitoneal chemotherapy has been introduced in order to improve outcomes for selected patients with peritoneal surface malignancies. Although survival benefits have been widely reported in the literature, this treatment is still not accepted worldwide because of the potential high incidence of postoperative complications. The aim of this study was to record the morbidity and mortality rates and to evaluate the associated risk factors. In our experience, cytoreductive surgery with hyperthermic intraperitoneal chemotherapy appeared as a safe and feasible procedure with good postoperative outcomes if performed in specialized centers. Further improvement of results could be achieved with better selection of patients.

**Abstract:**

Background: Cytoreductive surgery with hyperthermic intraperitoneal chemotherapy may significantly improve survival for selected patients with peritoneal surface malignancies, but it has always been criticized due to the high incidence of postoperative morbidity and mortality. Methods: Data were collected from nine Italian centers with peritoneal surface malignancies expertise within a collaborative group of the Italian Society of Surgical Oncology. Complications and mortality rates were recorded, and multivariate Cox analysis was used to identify risk factors. Results: The study included 2576 patients. The procedure was mostly performed for ovarian (27.4%) and colon cancer (22.4%). The median peritoneal cancer index was 13. Overall postoperative morbidity and mortality rates were 34% and 1.6%. A total of 232 (9%) patients required surgical reoperation. Multivariate regression logistic analysis identified the type of perfusion (*p* ≤ 0.0001), body mass index (*p* ≤ 0.0001), number of resections (*p* ≤ 0.0001) and colorectal resections (*p* ≤ 0.0001) as the strongest predictors of complications, whereas the number of resections (*p* ≤ 0.0001) and age (*p* = 0.01) were the strongest predictors of mortality. Conclusions: Cytoreductive surgery with hyperthermic intraperitoneal chemotherapy is a valuable option of treatment for selected patients with peritoneal carcinomatosis providing low postoperative morbidity and mortality rates, if performed in high-volume specialized centers.

## 1. Introduction

Cytoreductive surgery with hyperthermic intraperitoneal chemotherapy (CRS-HIPEC) has been increasingly considered an effective treatment for peritoneal surface malignancies (PSMs). Although it can significantly improve survival in selected patients with malignant mesothelioma and pseudomixoma peritoneii (PMP), the role in carcinomatosis originating from gastrointestinal and gynecological tumors is very promising but still under study [1,2,3,4,5,6]. The procedure has been widely criticized by the opponents due to the associated significant postoperative morbidity and mortality, thus preventing possibility of cure for many patients and clinical trials development. A systematic review reported in 2009 showed morbidity and mortality rates up to 52% and 17%, respectively [7]. Since the need for a steep learning curve including approximately 100 procedures each year has been clearly demonstrated [2,5,6,8,9], postoperative outcomes following CRS-HIPEC are similar to other major surgical procedures if performed in high-volume center [10,11,12,13,14]. A recent comparison with other high-risk surgical oncology procedures evaluating 34.114 patients showed that overall, 30-day mortality was lower in CRS/HIPEC (1.1%), compared with Whipple (2.5%), major hepatectomy (2.9–3.9%), esophagectomy (3.0%) [11]. The current morbidity and mortality rates vary from 22% to 45% and from 0.8% to 4%, respectively [1,2,3,4,5,6,11,12,13,14,15,16,17,18].

The purpose of this study was to evaluate the results of patients who underwent CRS-HIPEC for PSMs, specifically regarding postoperative complications and mortality rates, in order to identify possible risk factors.

## 2. Materials and Methods

A retrospective analysis of a prospectively maintained database of patients who underwent CRS-HIPEC for PSMs between 2000 and 2021 was conducted by the Italian Peritoneal Surface Malignancies Oncoteam of the Italian Society of Surgical Oncology (SICO). All the participating centers are referral centers certified by SICO for the surgical treatment of patients with peritoneal metastases.

Preoperative work-up included measurement of the serum tumor markers according to the different histological types and computed tomography (CT) scans in all patients. Esophagastroduodenoscopy and colonoscopy were performed in most cases, whereas magnetic resonance imaging (MRI), positron emission tomography (PET)/CT scans and other examinations were performed in selected cases. Some of the patients underwent preoperative laparoscopy in order to improve staging and selection. The extent of peritoneal involvement was assessed using the Peritoneal Cancer Index (PCI) and surgery was performed in accordance with Sugarbaker’s techniques [19]. Principles of surgery included the resection of each involved region and organ in order to achieve a complete macroscopic cytoreduction (Table 1).

HIPEC was performed immediately after cytoreduction using different protocols depending on each center’s preference, differing in the type of drug, concentrations, temperature, duration of the treatment and the technique of delivery. Three different types of administration were used: open, closed and semi-open. Briefly in the former, the abdominal wall is suspended up to a retractor and covered with a plastic sheet, thus creating an arena-like setup (“Coliseum” technique) and perfusion is performed under direct vision. In the closed technique, the abdominal wall is closed prior to HIPEC delivery. The semi-open technique is a cross between the other two, involving the use of different abdominal cavity expanders [20]. The chemotherapeutic agent was selected according to the primary tumor as well as the patient’s previous response to systemic chemotherapy, mainly including: cisplatin and doxorubicin for ovarian, gastric and mesothelioma cases, and mitomycin and oxaliplatin for colorectal, appendiceal and pseudomixoma cases. Two inflow and three outflow drains were used and left at the end of the procedure. The drugs were added to 4000 mL isotonic saline solution, the perfusate was heated to a temperature of 41–42.5 °C and circulated into the peritoneal cavity for 60–90 min, excluding oxaliplatin for 30 min. The dose of perfusion was reduced to minimize toxicity in some cases (e.g., elderly patients with poor performance status). Two intrabdominal thermometers (upper and lower abdomen) were used to monitor the temperature inside the peritoneal cavity during the infusion. The mean flow rate was 1200–2000 mL/min and global amount of perfusion approximately 4000 mL, depending on pathology and patient’s characteristics. At the end of the procedure, an abdominal washout was performed with 3L of crystalloids solution. Chest drains were selectively placed in case of diaphragmatic surgery. Following the procedure, most patients were admitted in the Intensive Care Unit for recovery until all vital signs were established. All patients received adequate pain control, perioperative antibiotic and venous thromboembolism prophylaxis with low molecular weight heparin. Administration of perioperative systemic chemotherapy followed international guidelines and varied in accordance with the type of cancer and patient medical history. Postoperative outcomes were categorized according to the National Cancer Institute’s Common Terminology Criteria for Adverse Events (NCI-CTCAE) [21]. A five-point scale to grade the severity of post-procedural adverse events was used. Clinical observation was the only required treatment for minor complications (grade I). Moderate complications required only minimal medical intervention (grade II). Severe complications required imaging-guided percutaneous or surgical drainage (including chest tube or therapeutic endoscopy) (grade III). Life-threatening complications requiring urgent intervention or intubation were graded IV. Deaths related to complications were graded V and calculated at 30, 60 and 90-day. Patients were regularly followed-up with blood tests and CT every 3 months for the first two years, every 6 months from years 3–5 and yearly thereafter or on demand, depending on the clinical status. For all outcomes, patients were censored from the date of surgery until death or last follow-up.

### Statistical Analysis

Demographic, clinical and perioperative variables were recorded for all patients. Descriptive statistics were reported as median (minimum and maximum values) or frequency (percentage) and used to summarize pertinent study information. The association between variables was tested by the Pearson Chi-Square test. The Odds Ratio (OR) and the confidence limits (CI95%) were estimated for the variables using the univariate logistic regression model. Significance was defined at the *p* < 0.05 level. A multivariate logistic regression model was also developed using stepwise regression (forward selection) with predictive variables which were significant in the univariate analyses. Enter limit and remove limit were *p* = 0.05 and *p* = 0.10, respectively. The SPSS (version 21.0; SPSS, Inc., Chicago, IL, USA) a licensed statistical program was used for all analyses.

## 3. Results

A total of 2576 patients who underwent CRS/HIPEC for PSMs in the period of study were analyzed. Baseline and perioperative findings for the entire cohort of patients are described in Table 2.

Median age was 58 years, and the vast majority were females (67.7%). Ovarian cancer represented the commonest indication (27.4%), followed by colon cancer (22.4%). Median PCI was 13, approximately two-thirds of patients (68.1%) underwent previous surgery, half of them (56.4%) previous chemotherapy and one third (32.7%) preoperative videolaparoscopy. The type of drugs for HIPEC varied according to the different histologies. Extended resections were required in most procedures, as depicted in Table 3. 

A total of 876 patients (34%) had single or combined complications, whereas 1700 patients (66%) presented no adverse events. Surgical type complications occurred in 567 cases (64.7%), medical type in 282 (32.2%) and mixed type in 27 (3.1%). Complications details are reported in Table 4.

Bleeding was the more frequent complication, accounting for 22.9% of cases. Taking into account the more serious one in cases with multiple complications, the CTCAE grading was as follows: 303 cases (34.6%) were qualified as grade I-II, 260 (29.7%) as grade III, 271 (30.9%) as grade IV and 42 patients (4.8%) as grade V (Table 5).

The 30-day, 60-day and 90-day mortality rate was 1.1% (25 patients), 0.2% (5 patients) and 0.3% (9 patients), respectively. The most common cause was respiratory failure, followed by bleeding. Surgical reoperation was required in 232 cases (9%), mostly due to bleeding (11%), perforation (6.9%) and anastomotic leakage (4.2%).

Among the variables identified for complications at univariate analysis (BMI, PCI, histology, diaphragmatic stripping and full thickness resection, splenectomy, hystero-adnexectomy, small bowel resection, appendectomy, liver resection, type of drug and perfusion, gastric and colorectal resections), only the type of perfusion (*p* ≤ 0.0001), BMI (*p* ≤ 0.0001), number of resections (*p* ≤ 0.0001) and colorectal resections (*p* ≤ 0.0001) were significant at multivariate analysis (Table 6).

Despite appendectomy emerging as a significant factor, we believe that it should be considered a misleading result, since it was always associated with all the strongest variables.

Among those variables associated with mortality at univariate analysis (age, PCI, right diaphragmatic stripping and diaphragmatic full thickness resection, splenectomy, small bowel resection and the number of resections), only the number of resections (*p* ≤ 0.0001) and age (*p* = 0.01) were significant at multivariate analysis (Table 7).

A Chi-Square test was then conducted on the association between the incidence of more frequent surgical complications and the most common histologies (Figure 1).

Bleeding was significantly less common in gastric cancer, anastomotic leakage and perforation higher in colon and gastric cancer, respectively (*p* = 0.014). Lastly, the test was also conducted on the association between the more severe grading of complications and the most common histologies (Chi-Square test) (Figure 2).

Gastric cancer showed a higher reintervention rate and a significantly higher mortality rate (*p* = 0.005).

## 4. Discussion

Aim of this study was to record the morbidity and mortality rates, as well as to evaluate the associated risk factors following CRS/HIPEC among 2576 patients collected from nine Italian centers with peritoneal surface malignancies expertise. In our experience, the procedure appeared safe and feasible, providing good postoperative outcomes. Overall postoperative morbidity and mortality rates were 34% and 1.6%. A total of 232 (9%) patients required surgical reoperation. Multivariate analysis identified the type of perfusion, BMI, number of resections and colorectal resections as the strongest predictors of complications, whereas the number of resections and age as the strongest predictors of mortality. Improvement of results could be achieved with better patient selection.

Despite increasing evidence on the therapeutic efficacy of CRS-HIPEC, some concerns still exist because of the potentially associated postoperative morbidity and mortality rates, due to the severe hemodynamic changes procedure-related. The occurrence of high-grade complications resulted an independently negative prognostic factor in several series [13,22,23,24,25]. The individual risk can be partly predicted by a variety of patients and operative factors, but the independent contribution of HIPEC seems relatively small [12,13,14].

Our overall postoperative morbidity rate was in line with that reported from high-volume centers [1,2,3,4,5,6,11,12,13,14,15,16,17,18,26,27,28,29,30]. The wide reported discrepancy in morbidity and mortality rates in the literature is mainly due to the different definitions and methods of classification used. In agreement with other authors, we believe that the NCI-CTCAE definition for types and grading of complications represents a legitimate metric of CRS-HIPEC quality, because it includes both medical and surgical conditions as well as many treatment-related complications and patients may die far beyond 30 days after surgery [1,3,4,5,16,26,27,28], as it occurred in approximately one third of our cases. Among the surgical complications, bleeding was the most frequent, followed by perforation and anastomotic leakage. Likewise, respiratory complications accounted for most of the medical type. We found a different incidence of surgical complications according to the histology. Overall incidence of bleeding was 22.9% and approximately half of the patients required reoperation, mostly because of oozing hemorrhage. This seemingly high rate is probably due both to the aggressive policy of surgery and the high number of ovarian, PN and mesothelioma cases requiring diaphragmatic surgery, as previously reported [31]. The occurrence of leakage was unsurprisingly increased in tumors requiring more anastomoses, such as colon cancer and PMP. The higher incidence of perforations in gastric cancer may be partly explained by the increased use of neoadjuvant chemotherapy in these patients. Aggressive surgery represents the cornerstone of treatment in CRS-HIPEC, since completeness of resection is the main prognostic factor for each tumor type [3,30,31]. Centralization in specialized centers is the key-factor in order to reduce the incidence of postoperative morbidity and mortality rates, but also to improve the perioperative management of patients. Careful selection of patients is also mandatory, especially in highly aggressive disease and the addiction of laparoscopy to the traditional imaging significantly improves the burden of disease estimation and prevents non-therapeutic laparotomies, basically excluding PMP cases and patients previously undergone multiple surgery [32,33,34]. The need for multiple resections, advanced age and poor performance status are the commonest independent risk factors for postoperative complications reported in the literature [13,15,18,35,36,37,38,39,40,41,42,43]. The need for multiple resections is related to disease extension and aggressive surgery, as reflected by higher PCI score. In our experience, it represented a significant predictor of complications only at univariate analysis, whereas the number of resections was significant at multivariate analysis for both morbidity and mortality rates. The degree of small bowel involvement may be more relevant than PCI itself to determine the potential for complete resection, since diffuse involvement of the small intestine and its mesentery represents the true key-point in determining resectability [33,34]. Despite the possible higher occurrence of complications, a massive disease diffusion should not be considered an exclusion criterion in neoplasms such as PMP and mesothelioma, since postoperative mortality is low and excellent survivals could be achieved even in advanced disease. In agreement with other series [15,16,25], colorectal resection was a significant predictor of complications at multivariate analysis. The result is not surprising, since this procedure was the most frequently performed and colorectal malignancies represented the second most frequent indication for CRS/HIPEC in our series. We do not have information as regards the American Society of Anesthesiologists score unfortunately, but BMI was significant predictor of complications at our multivariate analysis, and it may be considered a proxy for the assessment of poor performance status. The type of HIPEC administration was an unexpected factor influencing the occurrence of complications at our multivariate analysis, with the semi-open and the closed type being the most and the least safe, respectively. We have no statistical data, but some possible explanations could be given. In the closed technique, anastomoses may be considered at greater risk if performed before HIPEC delivery. Moreover, suction from drains at the end of the procedure may occasionally cause small bowel perforation. At the same time, the open technique allows better hemostasis control after HIPEC delivery. The optimal method has long been debated and prospective studies are lacking, but retrospective data suggest that the techniques are comparable in terms of intraoperative hemodynamic and postoperative morbidity [20,44,45]. Our result is presumably biased because HIPEC efficacy may be conditioned by several other parameters, such as type and concentrations of drug, carrier solution, volume and temperature of the perfusate, duration of the treatment and patient selection.

Our overall postoperative mortality rate also was in line with that reported from high-volume centers [1,2,3,4,5,6,11,12,13,14,15,16,17,18,26,27,28,29,30]. Respiratory complications and bleeding were the two most common causes accounting for 33% and 19% of deaths, respectively, as well as it happened for the occurrence of complications. Gastric cancer histology was significantly associated with the risk of mortality, clearly reflecting how accurate selection is particularly relevant for these patients. A definitive consensus for the cut-off point of PCI has not yet been established, but low score (≤6) is considered essential [46,47]. Coupled with the number of resections reflecting more complex procedures, age was the other significant predictor of mortality at our multivariate analysis. Advanced age has been traditionally associated with increased severe postoperative morbidity and mortality rates following complex procedures such as CRS/HIPEC, but frailty should be considered a stronger predictor than age alone for this heterogeneous group of patients with variable physiologic reserves [13,15,42,48,49,50].

The rate of reoperations was 9% in this study, in line with that reported in recent series, ranging from 9.8% to 16% [1,2,3,5,15,16,17,23,24,25,26,27,28,29]. In agreement with the literature, most common indications were represented by bleeding, perforation and leakage and the great majority usually occur within the first 12 days after surgery. Complications after CRS-HIPEC may occur far beyond the hospital length of stay and readmissions within 3 months after surgery may be required in up to 24% of patients [8,17,35,36,37,38,39,40,51,52,53]. Three main reasons have been identified: gastrointestinal, cardiovascular and miscellaneous. Bowel obstruction and intrabdominal abscess are the most common complications in the first group, which represents approximately two third of the total. Venous thrombembolic events constitute the most frequent cause of cardiovascular complication, whereas infections, renal failure and dehydration are the most common finding in the third category.

Aggressive cytoreduction, hyperthermia and length of operative time are responsible for a combination of hemodynamic, metabolic and hematologic disorders. Although these physiologic perturbations inherent to the procedure represent a significative perioperative challenge, to date no standard guidelines exist concerning intraoperative management. The use of goal-directed therapy enables to individually adjust the fluid therapy, avoiding over-hydration and ensuring hemodynamic stability of patients, resulting in lower morbidity and mortality rates, as well as length of hospital stay [54,55,56,57]. The procedure often includes complex, multivisceral resections and blood loss is significant in the great majority of patients. The lack of data on the number of blood transfusions is a major drawback of our study unfortunately. Since they may negatively impact perioperative and long-term outcomes of patients, sparing protocols are advisable [22,25,27,43,58,59,60]. 

One major limitation of this study is the heterogeneous cohort of patients as regards patient selection and management, as well as the length of the analysis time period. Additionally, inherent selection biases may be also present due to the retrospective nature of the data and the failure to evaluate other major risk factors. The incidence rate of Grade I-II complications is probably underestimated, but it is assumed that this does not have a significant impact on results. Nonetheless, the results come from one of the most important national registries, including only data from recognized high-volume centers. 

## 5. Conclusions

CRS-HIPEC can be considered an effective treatment for selected patients with PSM. Despite being a complex procedure, it is associated with low overall morbidity and mortality rates if performed in high-volume centers. Safety concerns should no longer be a deterrent to avoid the procedure. Further improvement of results could be obtained from a more accurate selection of cases and the standardization of protocols.

## Figures and Tables

**Figure 1 cancers-14-05824-f001:**
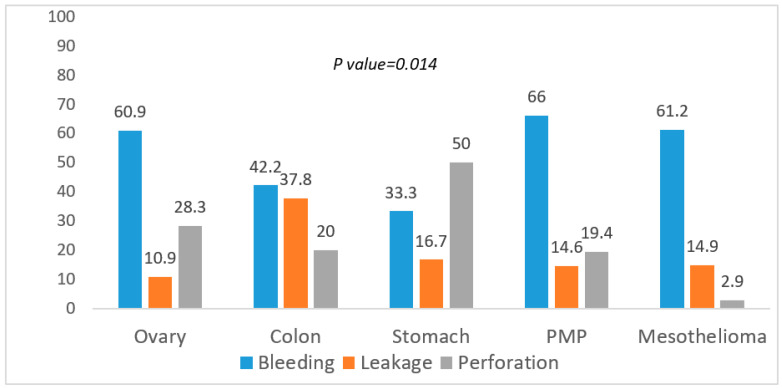
Association between surgical complications and histology.

**Figure 2 cancers-14-05824-f002:**
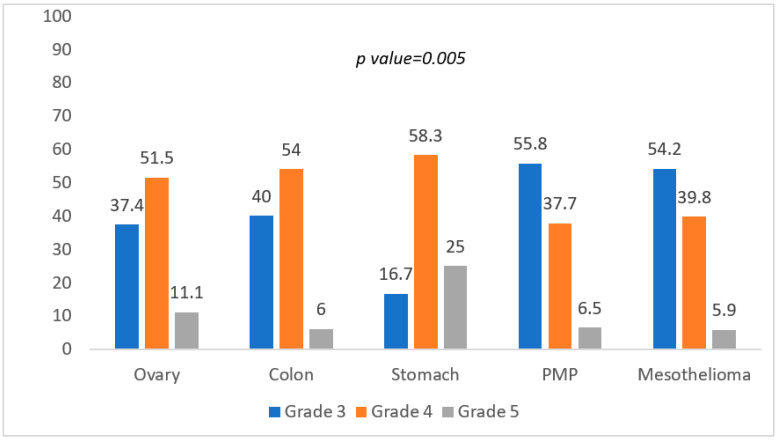
Association between grading of complications and histology.

**Table 1 cancers-14-05824-t001:** Listing of resected regions and organs.

Peritonectomy	Organs
Left parietal	Uterus and ovaries
Right parietal	Small bowel
Pelvic cavity	Rectosigmoid colon
Falciform and umbilical ligaments	Appendix
Hepatoduodenal ligament	Greater omentum
Glissonian capsule excision	Stomach
Epiploic retrocavity	Spleen
Mesenteric root excision	Liver
Treitz	Diaphragm
Diaphragmatic	Others (Gallbladder, Adrenal, Pancreas, Bladder, Ureter, Testicle)

**Table 2 cancers-14-05824-t002:** Baseline and perioperative features (total 2576 patients).

Median age, years (range)	58 (26–78)
Sex, n (%)	
Female	1719 (66.7)
Male	857 (33.3)
Median BMI, kg/m^2^, n (range)	25 (14–53)
Previous surgery, n (%)	1753 (68.1)
Previous VLS, n (%)	842 (32.7)
Previous CHT	1452 (56.4)
Histology, n (%)	
Ovary	706 (27.4)
Colon	577 (22.4)
PMP	505 (19.6)
Mesothelioma	352 (13.7)
Appendix	172 (6.7)
Stomach	135 (5.2)
Other	129 (5)
PCI, median (range)	13 (0–39)
Median resections, n (range)	5 (0–14)
HIPEC technique, n (%)	
Open	271 (10.5)
Closed	1668 (64.8)
Semi-open	637 (24.7)
Drugs type, n (%)	
Cisplatin	287 (11.1)
Doxorubicin + Cisplatin	860 (33.4)
Mytomicin	400 (15.5)
Mytomicin + Cisplatin	744 (28.9)
Oxaliplatin	285 (11.1)

BMI, body mass index; CHT, chemotherapy; VLS, videolaparoscopy; PMP, pseudomixoma peritoneii; PCI, Peritoneal Carcinosis Index; HIPEC, hyperthermic intraperitoneal chemotherapy.

**Table 3 cancers-14-05824-t003:** Surgical procedures N (%), (total 2576 patients).

Organs	
Hysteroadnexiectomies	862 (33.5)
Greater omentectomy	1849 (71.8)
Gastric resection	211 (8.2)
Colorectal resection	1917 (74.4)
Small bowel resection	656 (25.5)
Total colectomy	134 (5.2)
Splenectomy	1023 (39.7)
Right diaphragm FTR	1452 (56.4)
Left diaphragm FTR	1169 (45.4)
Pancreatic resection	44 (1.7)
Liver metastasectomy	119 (4.6)
Appendectomy	532 (23.7)
Gallbladder	1145 (44.6)
Other (ureter, bladder, testis, adrenal)	318 (12.3)
**Peritonectomies**	
Glissonian capsule	729 (28.3)
Mesenteric root	211 (8.2)
Treitz	266 (10.3)
Pelvic cavity	2083 (80.9)
Epiploic retrocavity	1316 (51.1)
Hepatoduodenal ligament	926 (35.9)
Falciform and umblical ligaments	1215 (47.2)
Right Parietal	1648 (64)
Left Parietal	1574 (61.1)
Right diaphragmatic S	1484 (57.6)
Left diaphragmatic S	1169 (45.4)

S, stripping; FTR, full-thickness resection.

**Table 4 cancers-14-05824-t004:** Complications, N (%) (total 952 cases).

Surgical Complications	
Bleeding	218 (22.9%)
Anastomotic leakage	70 (7.5%)
Perforation	89 (9.5%)
Abdominal wall eventration	33 (3.5%)
Bowel obstruction	26 (2.8%)
Ureteral leakage	20 (2.1%)
Bilio-pancreatic leakage	24 (2.5%)
Abscess	74 (7.8%)
**Medical Complications**	
Pancreatitis	28 (2.9%)
SSI	59 (6.2%)
Sepsis	69 (7.2%)
Portal thrombosis	5 (0.5%)
Pleural effusion	93 (9.8%)
Pneumonia	29 (3%)
Pulmonary embolism	33 (3.5%)
DVT	5 (0.5%)
Heart failure	9 (0.9%)
Acute Renal Failure	32 (3.3%)
Leukopenia	10 (1.%)
Neuropathy	10 (1.%)
Neurologic syndrome	10 (1.%)
TIA	2 (0.2%)
DIC	2 (0.2%)
MOF	2 (0.2%)

SSI, superficial site infection; DVT, deep vein thrombosis; MOF, multi organ failure; DIC, diffuse intravascular coagulation, TIA, transient ischemic attack.

**Table 5 cancers-14-05824-t005:** Complications grading, N (%), (total 876 patients).

Type of Complication	Grade I–II	Grade III	Grade IV	Grade V
Bleeding	30 (3.4%)	83 (9.4%)	97 (11%)	8 (0.9%)
Anastomotic leakage	7	20 (2.2%)	37 (4.2%)	5
Perforation	13 (1.5%)	11 (1.2%)	61 (6.9%)	3
Abdominal wall eventration	8	6	16 (1.8%)	-
Bowel obstruction	17 (1.9%)	1	-	-
Ureteral leakage	0	17 (1.9%)	8	-
Bilio-pancreatic leakage	1	14 (1.6%)	3	1
Abscess	26 (2.9%)	39 (4.4%)	10 (1.1%)	1
Pancreatitis	16 (1.8%)	3	8	-
SSI	51 (5.8%)	4	3	-
Sepsis	43 (4.9%)	7	-	4
Portal thrombosis	0	4	-	-
Pleural effusion	38 (4.3)	25 (2.8%)	1	-
Pneumonia	7	7	3	3
Pulmonary embolism	18 (2%)	-	7	11 (1.2%)
DVT	2	-	2	-
Heart failure	1	-	-	4
Acute Renal Failure	9 (0.8%)	10 (1.1%)	4	-
Leukopenia	5	1	5	-
Neuropathy	-	7	-	-
Neurologic syndrome	5	1	-	-
Transient ischemic attack	1	-	1	-
DIC	-	-	-	1
MOF	1	-	-	1

SSI, superficial site infection; DVT, deep vein thrombosis; MOF, multi organ failure; DIC, diffuse intravascular coagulation.

**Table 6 cancers-14-05824-t006:** Multivariate analysis for overall complications (total 952 cases).

Variables	Multivariate Analysis	
	OR (CI95%)	*p*
**Perfusion Type**	-	0.001
Semiopen vs. open	0.978 (0.681–1.404)	0.905
Closed vs. open	1.455 (1.058–2.001)	0.021
Semiopen vs. open	0.673 (0.532–0.851)	0.001
**Number of resections**	1.179 (1.133–1.227)	<0.0001
**Body Mass Index**	1.044 (1.022–1.067)	<0.0001
**Appendectomy** yes vs. no	1.422 (1.126–1.796)	0.003
**Colorectal reesctions** yes vs. no	1.526 (1.224–1.902)	<0.0001

**Table 7 cancers-14-05824-t007:** Multivariate analysis for grade V complications (total 952 cases).

Variables	Multivariate Analysis	
	OR (CI95%)	*p*
**Number of resections**	1.219 (1.091–1.361)	<0.0001
**Age**	1.089 (1.047–1.132)	<0.0001

## Data Availability

No new data were created or analyzed in this study. Data sharing is not applicable in this article.

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
