# Peer review of "Complications and Mortality Rate of Cytoreductive Surgery with Hyperthermic Intraperitoneal Chemotherapy: Italian Peritoneal Surface Malignancies Oncoteam Results Analysis"

_cancers, 2022, doi:10.3390/cancers14235824_

Round 1

Reviewer 1 Report

This is an impressive cohort of HIPEC patients treated in centers of excellence in Italy. Although retrospective, the absolute number of patients lends weight to the authors' data and conclusions. There are a few suggestions:

1. The rates of morbidity and mortality are acceptable but I would make an effort to compare these to other less experience centers using HIPEC as well as patients undergoing radical debulking surgery with out HIPEC.

2. Formatting of Table 3 needs to be fixed, may just be the pdf

3. I would expound upon the finding of perfusion type and its association with perioperative outcomes. I would ask the author to briefly explain each technique and speculate as to the reason for this difference. The other negative risk factors, age, BMI and # of resections all make sense. 

4. Bleeding post op seems high. Why? Are we capturing all patients that received blood or only those that went back to the OR for assessment. I would state this more clearly. And, if this is re-operation I would speculate as to why.

Finally, it would lend strength to the manuscript to compare those patients with poor outcomes to one of the several predictive models of surgical morbidity or HIPEC specific ones. This may help validate these models and identify patients whom may be at adverse risk 

Author Response

This is an impressive cohort of HIPEC patients treated in centers of excellence in Italy. Although retrospective, the absolute number of patients lends weight to the authors' data and conclusions. There are a few suggestions:

  1. The rates of morbidity and mortality are acceptable but I would make an effort to compare these to other less experience centers using HIPEC as well as patients undergoing radical debulking surgery with out HIPEC.

A couple of sentences concerning these topics have been added in the Introduction section, including a new reference.

  1. Formatting of Table 3 needs to be fixed, may just be the pdf

Table has been fixed, following the formatting error when uploaded to the system. 

  1. I would expound upon the finding of perfusion type and its association with perioperative outcomes. I would ask the author to briefly explain each technique and speculate as to the reason for this difference. The other negative risk factors, age, BMI and # of resections all make sense. 

Separated sentences have been inserted in the Methods and Discussion section, in order to briefly explain each technique and speculate on the different results. A new relevant reference has been also added.

  1. Bleeding post op seems high. Why? Are we capturing all patients that received blood or only those that went back to the OR for assessment. I would state this more clearly. And, if this is re-operation I would speculate as to why.

A new period has been inserted in the Results section, in order to both clarify results and speculate on the results, as requested. Moreover, a new relevant reference has been added.

  1. Finally, it would lend strength to the manuscript to compare those patients with poor outcomes to one of the several predictive models of surgical morbidity or HIPEC specific ones. This may help validate these models and identify patients whom may be at adverse risk 

This is an extremely interesting and current topic concerning major surgery. To the best of our knowledge, very few models predictor of complications and reamissions have been reported after CRS/HIPEC, but that was out our purpose. Moreover, the statistical significance would have been extremely low, due to the inherent limitations of our study, especially considering the multicentric nature and different pathologies inclusion.

Reviewer 2 Report

The authors present a retrospective review of a large series of patients who underwent CRS/HIPEC for PSM of various primaries. Majority of cases were performed for ovarian and colon cancer with median PCI of 13. This study demonstrated a morbidity and mortality rate similar to prior studies. Multivariate analysis demonstrated that perfusion type, BMI, number of resections, and colorectal resections predicted complications and number of resections and age predicted mortality. 

Broad/Specific Comments:

1. Unclear if this is due to original formatting or change in formatting when uploaded to the system, but some of the tables (particularly Table 4) have alignment issues that make them hard to read. 

2. I would recommend being more clear in the Materials/Methods section regarding patient selection. What time frame were these patients selected from and how many centers? I realize some of this information is in abstract, but should be delineated in the methods. 

3. Is there data regarding the operative time or operative start time? These could potentially impact the complication and mortality rate.

4. What was the completeness of cytoreduction score in these cases? Was this parameter considered in the multivariate analysis? 

Author Response

The authors present a retrospective review of a large series of patients who underwent CRS/HIPEC for PSM of various primaries. Majority of cases were performed for ovarian and colon cancer with median PCI of 13. This study demonstrated a morbidity and mortality rate similar to prior studies. Multivariate analysis demonstrated that perfusion type, BMI, number of resections, and colorectal resections predicted complications and number of resections and age predicted mortality. 

Broad/Specific Comments:

  1. Unclear if this is due to original formatting or change in formatting when uploaded to the system, but some of the tables (particularly Table 4) have alignment issues that make them hard to read. 

Table has been fixed, following the formatting error when uploaded to the system. 

  1. I would recommend being more clear in the Materials/Methods section regarding patient selection. What time frame were these patients selected from and how many centers? I realize some of this information is in abstract, but should be delineated in the methods. 

Requested data have been inserted at the beginnning of the Methods section

  1. Is there data regarding the operative time or operative start time? These could potentially impact the complication and mortality rate.

We are in agreement that operative time presumably impact the early postoperative results A mean value has been inserted in Table 1, but it was not included in the statistical analysis due to the extreme variability and complexity. It consitutes a potential bias of the study, as reported in the Limitations section.

  1. What was the completeness of cytoreduction score in these cases? Was this parameter considered in the multivariate analysis? 

Since only patients submitted to CC0-CC1 resection have been included in the study (as now highlighted at the beginning of the Results section), this parameter was not considered in the analysis